# Stochastic Three-Composite Convex Minimization

**Alp Yurtsever, Bằng Công Vũ, and Volkan Cevher**

Laboratory for Information and Inference Systems (LIONS)
École Polytechnique Fédérale de Lausanne, Switzerland
alp.yurtsever@epfl.ch, bang.vu@epfl.ch, volkan.cevher@epfl.ch

## Abstract

We propose a stochastic optimization method for the minimization of the sum of three convex functions, one of which has Lipschitz continuous gradient as well as restricted strong convexity. Our approach is most suitable in the setting where it is computationally advantageous to process smooth term in the decomposition with its stochastic gradient estimate and the other two functions separately with their proximal operators, such as doubly regularized empirical risk minimization problems. We prove the convergence characterization of the proposed algorithm in expectation under the standard assumptions for the stochastic gradient estimate of the smooth term. Our method operates in the primal space and can be considered as a stochastic extension of the three-operator splitting method. Numerical evidence supports the effectiveness of our method in real-world problems.

## 1 Introduction

We propose a stochastic optimization method for the three-composite minimization problem:

$$\underset{\mathbf{x}\in\mathbb{R}^d}{\text{minimize}}\ f(\mathbf{x}) + g(\mathbf{x}) + h(\mathbf{x}), \tag{1}$$

where $f : \mathbb{R}^d \to \mathbb{R}$ and $g : \mathbb{R}^d \to \mathbb{R}$ are proper, lower semicontinuous convex functions that admit tractable proximal operators, and $h : \mathbb{R}^d \to \mathbb{R}$ is a smooth function with restricted strong convexity. We assume that we have access to unbiased, stochastic estimates of the gradient of $h$ in the sequel, which is key to scale up optimization and to address streaming settings where data arrive in time.

Template (1) covers a large number of applications in machine learning, statistics, and signal processing by appropriately choosing the individual terms. Operator splitting methods are powerful in this setting, since they reduce the complex problem (1) into smaller subproblems. These algorithms are easy to implement, and they typically exhibit state-of-the-art performance.

To our knowledge, there is no operator splitting framework that can currently tackle template (1) using stochastic gradient of $h$ and the proximal operators of $f$ and $g$ separately, which is critical to the scalability of the methods. This paper specifically bridges this gap.

Our basic framework is closely related to the deterministic three operator splitting method proposed in [11], but we avoid the computation of the gradient $\nabla h$ and instead work with its unbiased estimates. We provide rigorous convergence guarantees for our approach and provide guidance in selecting the learning rate under different scenarios.

**Road map.** Section 2 introduces the basic optimization background. Section 3 then presents the main algorithm and provides its convergence characterization. Section 4 places our contributions in light of the existing work. Numerical evidence that illustrates our theory appears in Section 5. We relegate the technical proofs to the supplementary material.

## 2 Notation and background

This section recalls a few basic notions from the convex analysis and the probability theory, and presents the notation used in the rest of the paper. Throughout, $\Gamma_0(\mathbb{R}^d)$ denotes the set of all proper, lower semicontinuous convex functions from $\mathbb{R}^d$ to $[-\infty, +\infty]$, and $\langle \cdot \mid \cdot \rangle$ is the standard scalar product on $\mathbb{R}^d$ with its associated norm $\| \cdot \|$.

**Subdifferential.** The subdifferential of $f \in \Gamma_0(\mathbb{R}^d)$ at a point $\mathbf{x} \in \mathbb{R}^d$ is defined as

$$\partial f(\mathbf{x}) = \{\mathbf{u} \in \mathbb{R}^d \mid f(\mathbf{y}) - f(\mathbf{x}) \geq \langle \mathbf{y} - \mathbf{x} \mid \mathbf{u} \rangle, \forall \mathbf{y} \in \mathbb{R}^d\}.$$

We denote the domain of $\partial f$ as

$$\mathrm{dom}(\partial f) = \{\mathbf{x} \in \mathbb{R}^d \mid \partial f(\mathbf{x}) \neq \varnothing\}.$$

If $\partial f(\mathbf{x})$ is a singleton, then $f$ is a differentiable function, and $\partial f(\mathbf{x}) = \{\nabla f(\mathbf{x})\}$.

**Indicator function.** Given a nonempty subset $\mathcal{C}$ in $\mathbb{R}^d$, the indicator function of $\mathcal{C}$ is given by

$$\iota_{\mathcal{C}}(\mathbf{x}) = \begin{cases} 0 & \text{if } \mathbf{x} \in \mathcal{C}, \\ +\infty & \text{if } \mathbf{x} \notin \mathcal{C}. \end{cases} \tag{2}$$

**Proximal operator.** The proximal operator of a function $f \in \Gamma_0(\mathbb{R}^d)$ is defined as follows

$$\mathrm{prox}_f(\mathbf{x}) = \arg\min_{\boldsymbol{z} \in \mathbb{R}^d} \left\{ f(\boldsymbol{z}) + \frac{1}{2}\|\boldsymbol{z} - \mathbf{x}\|^2 \right\}. \tag{3}$$

Roughly speaking, the proximal operator is tractable when the computation of (3) is cheap. If $f$ is the indicator function of a nonempty, closed convex subset $\mathcal{C}$, its proximity operator is the projection operator on $\mathcal{C}$.

**Lipschitz continuos gradient.** A function $f \in \Gamma_0(\mathbb{R}^d)$ has Lipschitz continuous gradient with Lipschitz constant $L > 0$ (or simply $L$-Lipschitz), if

$$\|\nabla f(\mathbf{x}) - \nabla f(\mathbf{y})\| \leq L\|\mathbf{x} - \mathbf{y}\|, \qquad \forall \mathbf{x}, \mathbf{y} \in \mathbb{R}^d.$$

**Strong convexity.** A function $f \in \Gamma_0(\mathbb{R}^d)$ is called strongly convex with some parameter $\mu > 0$ (or simply $\mu$-strongly convex), if

$$\langle \boldsymbol{p} - \boldsymbol{q} \mid \mathbf{x} - \mathbf{y} \rangle \geq \mu\|\mathbf{x} - \mathbf{y}\|^2, \qquad \forall \mathbf{x}, \mathbf{y} \in \mathrm{dom}(\partial f), \ \forall \boldsymbol{p} \in \partial f(\mathbf{x}), \ \forall \boldsymbol{q} \in \partial f(\mathbf{y}).$$

**Solution set.** We denote optimum points of (1) by $\mathbf{x}^\star$, and the solution set by $\mathcal{X}^\star$:

$$\mathbf{x}^\star \in \mathcal{X}^\star = \{\mathbf{x} \in \mathbb{R}^d \mid \mathbf{0} \in \nabla h(\mathbf{x}) + \partial g(\mathbf{x}) + \partial f(\mathbf{x})\}.$$

Throughout this paper, we assume that $\mathcal{X}^\star$ is not empty.

**Restricted strong convexity.** A function $f \in \Gamma_0(\mathbb{R}^d)$ has restricted strong convexity with respect to a point $\mathbf{x}^\star$ in a set $\mathcal{M} \subset \mathrm{dom}(\partial f)$, with parameter $\mu > 0$, if

$$\langle \boldsymbol{p} - \boldsymbol{q} \mid \mathbf{x} - \mathbf{x}^\star \rangle \geq \mu\|\mathbf{x} - \mathbf{x}^\star\|^2, \qquad \forall \mathbf{x} \in \mathcal{M}, \ \forall \boldsymbol{p} \in \partial f(\mathbf{x}), \ \forall \boldsymbol{q} \in \partial f(\mathbf{x}^\star).$$

Let $(\Omega, \mathcal{F}, \mathsf{P})$ be a probability space. An $\mathbb{R}^d$-valued random variable is a measurable function $\mathbf{x} \colon \Omega \to \mathbb{R}^d$, where $\mathbb{R}^d$ is endowed with the Borel $\sigma$-algebra. We denote by $\sigma(\mathbf{x})$ the $\sigma$-field generated by $\mathbf{x}$. The expectation of a random variable $\mathbf{x}$ is denoted by $\mathbf{E}[\mathbf{x}]$. The conditional expectation of $\mathbf{x}$ given a $\sigma$-field $\mathcal{A} \subset \mathcal{F}$ is denoted by $\mathbf{E}[\mathbf{x}|\mathcal{A}]$. Given a random variable $\mathbf{y} \colon \Omega \to \mathbb{R}^d$, the conditional expectation of $\mathbf{x}$ given $\mathbf{y}$ is denoted by $\mathbf{E}[\mathbf{x}|\mathbf{y}]$. See [17] for more details on probability theory. An $\mathbb{R}^d$-valued random process is a sequence $(\mathbf{x}_n)_{n \in \mathbb{N}}$ of $\mathbb{R}^d$-valued random variables.

## 3 Stochastic three-composite minimization algorithm and its analysis

We present stochastic three-composite minimization method (S3CM) in Algorithm 1, for solving the three-composite template (1). Our approach combines the stochastic gradient of $h$, denoted as $\mathbf{r}$, and the proximal operators of $f$ and $g$ in essentially the same structrure as the three-operator splitting method [11, Algorithm 2]. Our technique is a nontrivial combination of the algorithmic framework of [11] with stochastic analysis.

**Algorithm 1** Stochastic three-composite minimization algorithm (S3CM)

---

*Input:* An initial point $\mathbf{x}_{f,0}$, a sequence of learning rates $(\gamma_n)_{n\in\mathbb{N}}$, and a sequence of squared integrable $\mathbb{R}^d$-valued stochastic gradient estimates $(\mathbf{r}_n)_{n\in\mathbb{N}}$.

*Initialization:*
$\quad \mathbf{x}_{g,0} = \text{prox}_{\gamma_0 g}(\mathbf{x}_{f,0})$
$\quad \mathbf{u}_{g,0} = \gamma_0^{-1}(\mathbf{x}_{f,0} - \mathbf{x}_{g,0})$

*Main loop:*
**for** $n = 0, 1, 2, \ldots$ **do**
$\quad \mathbf{x}_{g,n+1} = \text{prox}_{\gamma_n g}(\mathbf{x}_{f,n} + \gamma_n \mathbf{u}_{g,n})$
$\quad \mathbf{u}_{g,n+1} = \gamma_n^{-1}(\mathbf{x}_{f,n} - \mathbf{x}_{g,n+1}) + \mathbf{u}_{g,n}$
$\quad \mathbf{x}_{f,n+1} = \text{prox}_{\gamma_{n+1} f}(\mathbf{x}_{g,n+1} - \gamma_{n+1}\mathbf{u}_{g,n+1} - \gamma_{n+1}\mathbf{r}_{n+1})$
**end for**

*Output:* $\mathbf{x}_{g,n}$ as an approximation of an optimal solution $\mathbf{x}^\star$.

---

**Theorem 1** *Assume that $h$ is $\mu_h$-strongly convex and has $L$-Lipschitz continuous gradient. Further assume that $g$ is $\mu_g$-strongly convex, where we allow $\mu_g = 0$. Consider the following update rule for the learning rate:*

$$\gamma_{n+1} = \frac{-\gamma_n^2 \mu_h \eta + \sqrt{(\gamma_n^2 \mu_h \eta)^2 + (1 + 2\gamma_n \mu_g)\gamma_n^2}}{1 + 2\gamma_n \mu_g}, \qquad \text{for some } \gamma_0 > 0 \text{ and } \eta \in ]0,1[.$$

*Define $\mathcal{F}_n = \sigma(\mathbf{x}_{f,k})_{0\le k\le n}$, and suppose that the following conditions hold for every $n \in \mathbb{N}$:*

1. *$\mathbf{E}[\mathbf{r}_{n+1}|\mathcal{F}_n] = \nabla h(\mathbf{x}_{g,n+1})$ almost surely,*

2. *There exists $c \in [0, +\infty[$ and $t \in \mathbb{R}$, that satisfies $\sum_{k=0}^{n} \mathbf{E}[\|\mathbf{r}_k - \nabla h(\mathbf{x}_{g,k})\|^2] \le cn^t$.*

*Then, the iterates of S3CM satisfy*
$$\mathbf{E}[\|\mathbf{x}_{g,n} - \mathbf{x}^\star\|^2] = \mathcal{O}(1/n^2) + \mathcal{O}(1/n^{2-t}). \tag{4}$$

**Remark 1** The variance condition of the stochastic gradient estimates in the theorems above is satisfied when $\mathbf{E}[\|\mathbf{r}_n - \nabla h(\mathbf{x}_{g,n})\|^2] \le c$ for all $n \in \mathbb{N}$ and for some constant $c \in [0, +\infty[$. See [15, 22, 26] for details.

**Remark 2** When $\mathbf{r}_n = \nabla h(\mathbf{x}_n)$, S3CM reduces to the deterministic three-operator splitting scheme [11, Algorithm 2] and we recover the convergence rate $\mathcal{O}(1/n^2)$ as in [11]. When $g$ is zero, S3CM reduces to the standard stochastic proximal point algorithm [2, 13, 26].

**Remark 3** Learning rate sequence $(\gamma_n)_{n\in\mathbb{N}}$ in Theorem 1 depends on the strong convexity parameter $\mu_h$, which may not be available a priori. Our next result avoids the explicit reliance on the strong convexity parameter, while providing essentially the same convergence rate.

**Theorem 2** *Assume that $h$ is $\mu_h$-strongly convex and has $L$-Lipschitz continuous gradient. Consider a positive decreasing learning rate sequence $\gamma_n = \Theta(1/n^\alpha)$ for some $\alpha \in ]0,1]$, and denote $\beta = \lim_{n\to\infty} 2\mu_h n^\alpha \gamma_n$.*

*Define $\mathcal{F}_n = \sigma(\mathbf{x}_{f,k})_{0\le k\le n}$, and suppose that the following conditions hold for every $n \in \mathbb{N}$:*

1. *$\mathbf{E}[\mathbf{r}_{n+1}|\mathcal{F}_n] = \nabla h(\mathbf{x}_{g,n+1})$ almost surely,*

2. *$\mathbf{E}[\|\mathbf{r}_n - \nabla h(\mathbf{x}_{g,n})\|^2]$ is uniformly bounded by some positive constant.*

3. *$\mathbf{E}[\|\mathbf{u}_{g,n} - \mathbf{x}^\star\|^2]$ is uniformly bounded by some positive constant.*

*Then, the iterates of S3CM satisfy*
$$\mathbf{E}[\|\mathbf{x}_{g,n} - \mathbf{x}^\star\|^2] = \begin{cases} \mathcal{O}(1/n^\alpha) & \text{if } 0 < \alpha < 1 \\ \mathcal{O}(1/n^\beta) & \text{if } \alpha = 1, \text{ and } \beta < 1 \\ \mathcal{O}((\log n)/n) & \text{if } \alpha = 1, \text{ and } \beta = 1, \\ \mathcal{O}(1/n) & \text{if } \alpha = 1, \text{ and } \beta > 1. \end{cases}$$

*Proof outline.* We consider the proof of three-operator splitting method as a baseline, and we use the stochastic fixed point theory to derive the convergence of the iterates via the stochastic Fejér monotone sequence. See the supplement for the complete proof.

**Remark 4** Note that $\mathbf{u}_{g,n} \in \partial g(\mathbf{x}_{g,n})$. Hence, we can replace condition 3 in Theorem 2 with the bounded subgradient assumption: $\|\boldsymbol{p}\| \leq c, \forall \boldsymbol{p} \in \partial g(\mathbf{x}_{g,n})$, for some positive constant $c$.

**Remark 5 (Restricted strong convexity)** Let $\mathcal{M}$ be a subset of $\mathbb{R}^d$ that contains $(\mathbf{x}_{g,n})_{n \in \mathbb{N}}$ and $\mathbf{x}^\star$. Suppose that $h$ has restricted strong convexity on $\mathcal{M}$ with parameter $\mu_h$. Then, Theorems 1 and 2 still hold. An example role of the restricted strong convexity assumption on algorithmic convergence can be found in [1, 21].

**Remark 6 (Extension to arbitrary number of non-smooth terms.)** Using the product space technique [5, Section 6.1], S3CM can be applied to composite problems with arbitrary number of non-smooth terms:

$$\underset{\mathbf{x} \in \mathbb{R}^d}{\text{minimize}} \sum_{i=1}^{m} f_i(\mathbf{x}) + h(\mathbf{x}),$$

where $f_i : \mathbb{R}^d \to \mathbb{R}$ are proper, lower semicontinuous convex functions, and $h : \mathbb{R}^d \to \mathbb{R}$ is a smooth function with restricted strong convexity. We present this variant in Algorithm 2. Theorems 1 and 2 hold for this variant, replacing $\mathbf{x}_{g,n}$ by $\overline{\mathbf{x}}_n$, and $\mathbf{u}_{g,n}$ by $\mathbf{u}_{i,n}$ for $i = 1, 2, \ldots, m$.

---

**Algorithm 2** Stochastic m(ulti)-composite minimization algorithm (SmCM)

---

*Input:* Initial points $\{\mathbf{x}_{f_1,0}, \mathbf{x}_{f_2,0}, \ldots, \mathbf{x}_{f_m,0}\}$, a sequence of learning rates $(\gamma_n)_{n \in \mathbb{N}}$, and a sequence of squared integrable $\mathbb{R}^d$-valued stochastic gradient estimates $(\mathbf{r}_n)_{n \in \mathbb{N}}$

*Initialization:*
$\overline{\mathbf{x}}_0 = m^{-1} \sum_{i=1}^{m} \mathbf{x}_{f_i,0}$
**for** i=1,2,...,m **do**
 $\quad \mathbf{u}_{i,0} = \gamma_0^{-1}(\mathbf{x}_{f_i,0} - \overline{\mathbf{x}}_0)$
**end for**

*Main loop:*
**for** $n = 0, 1, 2, \ldots$ **do**
 $\quad \overline{\mathbf{x}}_{n+1} = m^{-1} \sum_{i=1}^{m}(\mathbf{x}_{f_i,n} + \gamma_n \mathbf{u}_{i,n})$
 $\quad$ **for** i=1,2,...,m **do**
 $\quad\quad \mathbf{u}_{i,n+1} = \gamma_n^{-1}(\mathbf{x}_{f_i,n} - \overline{\mathbf{x}}_{n+1}) + \mathbf{u}_{i,n}$
 $\quad\quad \mathbf{x}_{f_i,n+1} = \text{prox}_{\gamma_{n+1} m f_i}(\overline{\mathbf{x}}_{n+1} - \gamma_{n+1}\mathbf{u}_{i,n+1} - \gamma_{n+1}\mathbf{r}_{n+1})$
 $\quad$ **end for**
**end for**

*Output:* $\overline{\mathbf{x}}_n$ as an approximation of an optimal solution $\mathbf{x}^\star$.

---

**Remark 7** With a proper learning rate, S3CM still converges even if $h$ is not (restricted) strongly convex under mild assumptions. Suppose that $h$ has $L$-Lipschitz continuous gradient. Set the learning rate such that $\varepsilon \leq \gamma_n \equiv \gamma \leq \alpha(2L^{-1} - \varepsilon)$, for some $\alpha$ and $\varepsilon$ in $]0, 1[$. Define $\mathcal{F}_n = \sigma(\mathbf{x}_{f,k})_{0 \leq k \leq n}$, and suppose that the following conditions hold for every $n \in \mathbb{N}$:

1. $\mathbf{E}[\mathbf{r}_{n+1}|\mathcal{F}_n] = \nabla \boldsymbol{h}(\mathbf{x}_{g,n+1})$ almost surely.

2. $\sum_{n \in \mathbb{N}} \mathbf{E}[\|\mathbf{r}_{n+1} - \nabla \boldsymbol{h}(\mathbf{x}_{g,n+1})\|^2 | \mathcal{F}_n] < +\infty$ almost surely.

Then, $(\mathbf{x}_{g,n})_{n \in \mathbb{N}}$ converges to a $\mathcal{X}^\star$-valued random vector almost surely. See [7] for details.

**Remark 8** All the results above hold for any separable Hilbert space, except that the strong convergence in Remark 7 is replaced by weak convergence. Note however that extending Remark 7 to variable metric setting as in [10, 27] is an open problem.

# 4    Contributions in the light of prior work

Recent algorithms in the operator splitting, such as generalized forward-backward splitting [24], forward-Douglas-Rachford splitting [5], and the three-operator splitting [11], apply to our problem template (1). These key results, however, are in the deterministic setting.

*Our basic framework can be viewed as a combination of the three-operator splitting method in [11] with the stochastic analysis.*

The idea of using unbiased estimates of the gradient dates back to [25]. Recent developments of this idea can be viewed as proximal based methods for solving the generic composite convex minimization template with a single non-smooth term [2, 9, 12, 13, 15, 16, 19, 26, 23]. This generic form arises naturally in regularized or constrained composite problems [3, 13, 20], where the smooth term typically encodes the data fidelity. These methods require the evaluation of the joint prox of $f$ and $g$ when applied to the three-composite template (1).

Unfortunately, evaluation of the joint prox is arguably more expensive compared to the individual prox operators. To make comparison stark, consider the simple example where $f$ and $g$ are indicator functions for two convex sets. Even if the projection onto the individual sets are easy to compute, projection onto the intersection of these sets can be challenging.

Related literature also contains algorithms that solve some specific instances of template (1). To point out a few, random averaging projection method [28] handles multiple constraints simultaneously but cannot deal with regularizers. On the other hand, accelerated stochastic gradient descent with proximal average [29] can handle multiple regularizers simultaneously, but the algorithm imposes a Lipschitz condition on regularizers, and hence, it cannot deal with constraints.

To our knowledge, our method is the first operator splitting framework that can tackle optimization template (1) using the stochastic gradient estimate of $h$ and the proximal operators of $f$ and $g$ separately, without any restriction on the non-smooth parts except that their subdifferentials are maximally monotone. When h is strongly convex, under mild assumptions, and with a proper learning rate, our algorithm converges with $\mathcal{O}(1/n)$ rate, which is optimal for the stochastic methods under strong convexity assumption for this problem class.

# 5    Numerical experiments

We present numerical evidence to assess the theoretical convergence guarantees of the proposed algorithm. We provide two numerical examples from Markowitz portfolio optimization and support vector machines.

As a baseline, we use the deterministic three-operator splitting method [11]. Even though the random averaging projection method proposed in [28] does not apply to our template (1) with its all generality, it does for the specific applications that we present below. In our numerical tests, however, we observed that this method exhibits essentially the same convergence behavior as ours when used with the same learning rate sequence. For the clarity of the presentation, we omit this method in our results.

## 5.1    Portfolio optimization

Traditional Markowitz portfolio optimization aims to reduce risk by minimizing the variance for a given expected return. Mathematically, we can formulate this as a convex optimization problem [6]:

$$\underset{\mathbf{x} \in \mathbb{R}^d}{\text{minimize}} \quad \mathbf{E}\left[|a_i^T \mathbf{x} - b|^2\right] \quad \text{subject to} \quad \mathbf{x} \in \Delta, \quad a_{av}^T \mathbf{x} \geq b,$$

where $\Delta$ is the standard simplex for portfolios with no-short positions or a simple sum constraint, $a_{av} = \mathbf{E}[a_i]$ is the average returns for each asset that is assumed to be known (or estimated), and $b$ encodes a minimum desired return.

This problem has a streaming nature where new data points arrive in time. Hence, we typically do not have access to the whole dataset, and the stochastic setting is more favorable. For implementation,

we replace the expectation with the empirical sample average:

$$\underset{\mathbf{x}\in\mathbb{R}^d}{\text{minimize}} \quad \frac{1}{p}\sum_{i=1}^{p}(\boldsymbol{a}_i^T\mathbf{x}-b)^2 \quad \text{subject to} \quad \mathbf{x}\in\Delta, \quad \boldsymbol{a}_{av}^T\mathbf{x}\geq b. \tag{5}$$

This problem fits into our optimization template (1) by setting

$$h(\mathbf{x}) = \frac{1}{p}\sum_{i=1}^{p}(\boldsymbol{a}_i^T\mathbf{x}-b)^2, \quad g(\mathbf{x}) = \iota_\Delta(\mathbf{x}), \quad \text{and} \quad f(\mathbf{x}) = \iota_{\{\mathbf{x} \mid \boldsymbol{a}_{av}^T\mathbf{x}\geq b\}}(\mathbf{x}).$$

We compute the unbiased estimates of the gradient by $\mathbf{r}_n = 2(\boldsymbol{a}_{i_n}^T\mathbf{x}-b)\boldsymbol{a}_{i_n}$, where index $i_n$ is chosen uniformly random.

We use 5 different real portfolio datasets: Dow Jones industrial average (DJIA, with 30 stocks for 507 days), New York stock exchange (NYSE, with 36 stocks for 5651 days), Standard & Poor's 500 (SP500, with 25 stocks for 1276 days), Toronto stock exchange (TSE, with 88 stocks for 1258 days) that are also considered in [4]; and one dataset by Fama and French (FF100, 100 portfolios formed on size and book-to-market, 23,647 days) that is commonly used in financial literature, e.g., [6, 14]. We impute the missing data in FF100 using nearest-neighbor method with Euclidean distance.

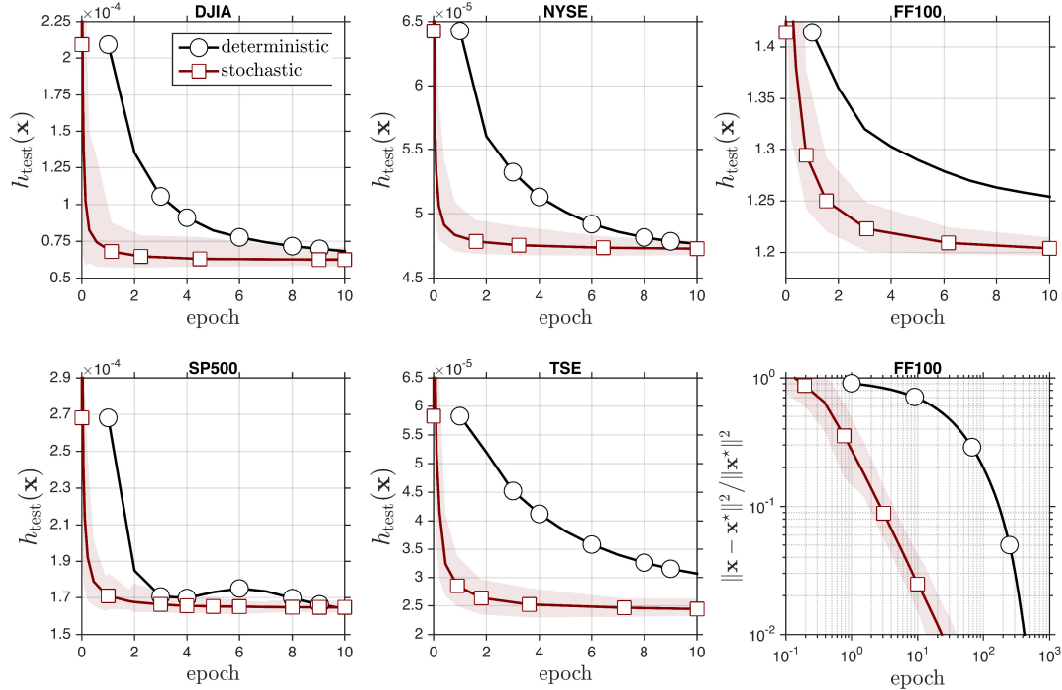

Figure 1: Comparison of the deterministic three-operators splitting method [11, Algorithm 2] and our stochastic three-composite minimization method (S3CM) for Markowitz portfolio optimization (5). Results are averaged over 100 Monte-Carlo simulations, and the boundaries of the shaded area are the best and worst instances.

For the deterministic algorithm, we set $\eta = 0.1$. We evaluate the Lipschitz constant $L$ and the strong convexity parameter $\mu_h$ to determine the step-size. For the stochastic algorithm, we do not have access to the whole data, so we cannot compute these parameter. Hence, we adopt the learning rate sequence defined in Theorem 2. We simply use $\gamma_n = \gamma_0/(n+1)$ with $\gamma_0 = 1$ for FF100, and $\gamma_0 = 10^3$ for others.[1] We start both algorithms from the zero vector.

We split all the datasets into test (10%) and train (90%) partitions randomly. We set the desired return as the average return over all assets in the training set, $b = \text{mean}(\boldsymbol{a}_{av})$. Other $b$ values exhibit qualitatively similar behavior.

The results of this experiment are compiled in Figure 1. We compute the objective function over the datapoints in the test partition, $h_{\text{test}}$. We compare our algorithm against the deterministic three-operator splitting method [11, Algorithm 2]. Since we seek statistical solutions, we compare the algorithms to achieve low to medium accuracy. [11] provides other variants of the deterministic algorithm, including two ergodic averaging schemes that feature improved theoretical rate of convergence. However, these variants performed worse in practice than the original method, and are omitted.

Solid lines in Figure 1 present the average results over 100 Monte-Carlo simulations, and the boundaries of the shaded area are the best and worst instances. We also assess empirical evidence of the $\mathcal{O}(1/n)$ convergence rate guaranteed in Theorem 2, by presenting squared relative distance to the optimum solution for FF100 dataset. Here, we approximate the ground truth by solving the problem to high accuracy with the deterministic algorithm for $10^5$ iterations.

## 5.2 Nonlinear support vector machines classification

This section demonstrates S3CM on a support vector machines (SVM) for binary classification problem. We are given a training set $\mathcal{A} = \{\boldsymbol{a}_1, \boldsymbol{a}_2, \ldots, \boldsymbol{a}_d\}$ and the corresponding class labels $\{b_1, b_2, \ldots, b_d\}$, where $\boldsymbol{a}_i \in \mathbb{R}^p$ and $b_i \in \{-1, 1\}$. The goal is to build a model that assigns new examples into one class or the other correctly.

As common in practice, we solve the dual soft-margin SVM formulation:

$$\underset{\mathbf{x} \in \mathbb{R}^d}{\text{minimize}} \quad \frac{1}{2} \sum_{i=1}^{d} \sum_{j=1}^{d} K(\boldsymbol{a}_i, \boldsymbol{a}_j) b_i b_j x_i x_j - \sum_{i=1}^{d} x_i \quad \text{subject to} \quad \mathbf{x} \in [0, C]^d, \quad \boldsymbol{b}^T \mathbf{x} = 0,$$

where $C \in [0, +\infty[$ is the penalty parameter and $K : \mathbb{R}^p \times \mathbb{R}^p \to \mathbb{R}$ is a kernel function. In our example we use the Gaussian kernel given by $K_\sigma(\boldsymbol{a}_i, \boldsymbol{a}_j) = \exp(-\sigma \|\boldsymbol{a}_i - \boldsymbol{a}_j\|^2)$ for some $\sigma > 0$.

Define symmetric positive semidefinite matrix $\boldsymbol{M} \in \mathbb{R}^{d \times d}$ with entries $M_{ij} = K_\sigma(\boldsymbol{a}_i, \boldsymbol{a}_j) b_i b_j$. Then the problem takes the form

$$\underset{\mathbf{x} \in \mathbb{R}^d}{\text{minimize}} \quad \frac{1}{2} \mathbf{x}^T \boldsymbol{M} \mathbf{x} - \sum_{i=1}^{d} x_i \quad \text{subject to} \quad \mathbf{x} \in [0, C]^d, \quad \boldsymbol{b}^T \mathbf{x} = 0. \tag{6}$$

This problem fits into three-composite optimization template (1) with

$$h(\mathbf{x}) = \frac{1}{2} \mathbf{x}^T \boldsymbol{M} \mathbf{x} - \sum_{i=1}^{d} x_i, \quad g(\mathbf{x}) = \iota_{[0,C]^d}(\mathbf{x}), \quad \text{and} \quad f(\mathbf{x}) = \iota_{\{\mathbf{x} \mid \boldsymbol{b}^T \mathbf{x} = 0\}}(\mathbf{x}).$$

One can solve this problem using three-operator splitting method [11, Algorithm 1]. Note that $\text{prox}_f$ and $\text{prox}_g$, which are projections onto the corresponding constraint sets, incur $\mathcal{O}(d)$ computational cost, whereas the cost of computing the gradient is $\mathcal{O}(d^2)$.

To compute an unbiased gradient estimate, we choose an index $i_n$ uniformly random, and we form $\mathbf{r}_n = d \boldsymbol{M}_{i_n} x_{i_n} - \mathbf{1}$. Here $\boldsymbol{M}_{i_n}$ denotes $i_n^{th}$ column of matrix $\boldsymbol{M}$, and $\mathbf{1}$ represents the vector of ones. We can compute $\mathbf{r}_n$ in $\mathcal{O}(d)$ computations, hence each iteration of S3CM costs an order cheaper compared to deterministic algorithm.

We use UCI machine learning dataset "a1a", with $d = 1605$ datapoints and $p = 123$ features [8, 18]. Note that our goal here is to demonstrate the optimization performance of our algorithm for a real world problem, rather than competing the prediction quality of the best engineered solvers. Hence, to keep experiments simple, we fix problem parameters $C = 1$ and $\sigma = 2^{-2}$, and we focus on the effects of algorithmic parameters on the convergence behavior.

Since $p < d$, $\boldsymbol{M}$ is rank deficient and $h$ is not strongly convex. Nevertheless we use S3CM with the learning rate $\gamma_n = \gamma_0/(n+1)$ for various values of $\gamma_0$. We observe $\mathcal{O}(1/n)$ empirical convergence rate on the squared relative error for large enough $\gamma_0$, which is guaranteed under restricted strong convexity assumption. See Figure 2 for the results.

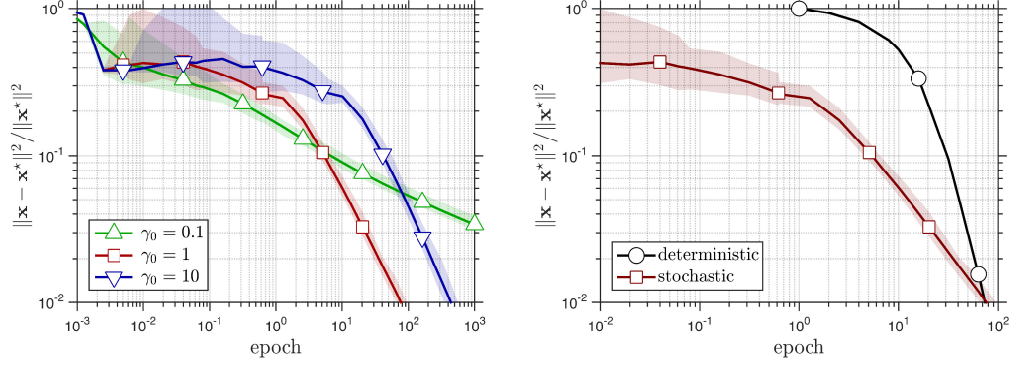

Figure 2: *[Left]* Convergence of S3CM in the squared relative error with learning rate $\gamma_n = \gamma_0/(n+1)$. *[Right]* Comparison of the deterministic three-operators splitting method [11, Algorithm 1] and S3CM with $\gamma_0 = 1$ for SVM classification problem. Results are averaged over 100 Monte-Carlo simulations. Boundaries of the shaded area are the best and worst instances.

### Acknowledgments

This work was supported in part by ERC Future Proof, SNF 200021-146750, SNF CRSII2-147633, and NCCR-Marvel.

## Footnotes

[1]Note that a fine-tuned learning rate with a more complex definition can improve the empirical performance, e.g., $\gamma_n = \gamma_0/(n+\zeta)$ for some positive constants $\gamma_0$ and $\zeta$.

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
