[Supplementary Material · YVC2016_S3CM_supp.pdf]

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

## Appendix: Proof of the main result

In this supplement, we provide the proofs of Theorem 1 and Theorem 2.

**Proof of Theorem 1**. For every $n \in \mathbb{N}$, we have

$$
\begin{cases}
\mathbf{u}_{f,n} = \gamma_n^{-1}(\mathbf{x}_{g,n} - \mathbf{x}_{f,n}) - (\mathbf{u}_{g,n} + \mathbf{r}_n) \in \partial f(\mathbf{x}_{f,n}) \\
\mathbf{u}_{g,n} \in \partial g(\mathbf{x}_{g,n}) \\
\gamma_n(\mathbf{u}_{g,n+1} - \mathbf{u}_{g,n}) = \mathbf{x}_{f,n} - \mathbf{x}_{g,n+1} \\
\gamma_n(\mathbf{u}_{f,n} + \mathbf{u}_{g,n} + \mathbf{r}_n) = \mathbf{x}_{g,n} - \mathbf{x}_{f,n} \\
\gamma_n(\mathbf{u}_{g,n+1} + \mathbf{u}_{f,n} + \mathbf{r}_n) = \mathbf{x}_{g,n} - \mathbf{x}_{g,n+1}.
\end{cases}
$$

Now, let us define

$$
\begin{cases}
\chi_n &= 2\gamma_n \langle \mathbf{x}_{f,n} - \mathbf{x}^\star \mid \mathbf{u}_{f,n} + \mathbf{r}_n \rangle + 2\gamma_n \langle \mathbf{x}_{g,n+1} - \mathbf{x}^\star \mid \mathbf{u}_{g,n+1} \rangle \\
\chi_{1,n} &= 2 \langle \mathbf{x}_{g,n+1} - \mathbf{x}^\star \mid \mathbf{x}_{g,n} - \mathbf{x}_{g,n+1} \rangle \\
\chi_{2,n} &= 2 \langle \mathbf{x}_{f,n} - \mathbf{x}_{g,n+1} \mid \mathbf{x}_{g,n} - \mathbf{x}_{f,n} \rangle \\
\chi_{3,n} &= 2\gamma_n \langle \mathbf{x}_{g,n+1} - \mathbf{x}_{f,n} \mid \mathbf{u}_{g,n} - \mathbf{u}_g^\star \rangle = 2\gamma_n^2 \langle \mathbf{u}_{g,n} - \mathbf{u}_{g,n+1} \mid \mathbf{u}_{g,n} - \mathbf{u}_g^\star \rangle \\
\chi_{4,n} &= 2\gamma_n \langle \mathbf{x}_{g,n+1} - \mathbf{x}_{f,n} \mid \mathbf{u}_g^\star \rangle,
\end{cases}
$$

where $\mathbf{u}_g^\star \in \partial g(\mathbf{x}^\star)$. Then, by simple calculations we get

$$
\begin{cases}
\chi_{2,n} &= \|\mathbf{x}_{g,n} - \mathbf{x}_{g,n+1}\|^2 - \|\mathbf{x}_{f,n} - \mathbf{x}_{g,n+1}\|^2 - \|\mathbf{x}_{g,n} - \mathbf{x}_{f,n}\|^2 \\
\chi_{1,n} &= \|\mathbf{x}_{g,n} - \mathbf{x}^\star\|^2 - \|\mathbf{x}_{g,n+1} - \mathbf{x}^\star\|^2 - \|\mathbf{x}_{g,n} - \mathbf{x}_{g,n+1}\|^2 \\
\chi_{3,n} &= \gamma_n^2 \|\mathbf{u}_{g,n+1} - \mathbf{u}_{g,n}\|^2 + \gamma_n^2 \|\mathbf{u}_{g,n} - \mathbf{u}_g^\star\|^2 - \gamma_n^2 \|\mathbf{u}_{g,n+1} - \mathbf{u}_g^\star\|^2 \\
&= \gamma_n^2 \|\mathbf{u}_{g,n} - \mathbf{u}_g^\star\|^2 - \gamma_n^2 \|\mathbf{u}_{g,n+1} - \mathbf{u}_g^\star\|^2 + \|\mathbf{x}_{f,n} - \mathbf{x}_{g,n+1}\|^2.
\end{cases}
\tag{7}
$$

Furthermore, for every $n \in \mathbb{N}$, we can express $\chi_n$ as follows:

$$
\begin{aligned}
\chi_n &= 2\gamma_n \langle \mathbf{x}_{f,n} - \mathbf{x}_{g,n+1} \mid \mathbf{u}_{f,n} + \mathbf{r}_n \rangle + 2\gamma_n \langle \mathbf{x}_{g,n+1} - \mathbf{x}^\star \mid \mathbf{u}_{g,n+1} + \mathbf{u}_{f,n} + \mathbf{r}_n \rangle \\
&= \chi_{1,n} + 2\gamma_n \langle \mathbf{x}_{f,n} - \mathbf{x}_{g,n+1} \mid \mathbf{u}_{f,n} + \mathbf{r}_n \rangle \\
&= \chi_{1,n} + 2\gamma_n \big( \langle \mathbf{x}_{f,n} - \mathbf{x}_{g,n+1} \mid \mathbf{u}_{f,n} + \mathbf{r}_n + \mathbf{u}_{g,n} \rangle - \langle \mathbf{x}_{f,n} - \mathbf{x}_{g,n+1} \mid \mathbf{u}_{g,n} \rangle \big) \\
&= \chi_{1,n} + \chi_{2,n} + 2\gamma_n \langle \mathbf{x}_{g,n+1} - \mathbf{x}_{f,n} \mid \mathbf{u}_{g,n} \rangle \\
&= \chi_{1,n} + \chi_{2,n} + 2\gamma_n \langle \mathbf{x}_{g,n+1} - \mathbf{x}_{f,n} \mid \mathbf{u}_g^\star \rangle + 2\gamma_n \langle \mathbf{x}_{g,n+1} - \mathbf{x}_{f,n} \mid \mathbf{u}_{g,n} - \mathbf{u}_g^\star \rangle \\
&= \chi_{1,n} + \chi_{2,n} + \chi_{3,n} + \chi_{4,n}.
\end{aligned}
$$

Now, summing the equalities in (7), we obtain,

$$
\begin{aligned}
\chi_n = \gamma_n^2 \big( \|\mathbf{u}_{g,n} - \mathbf{u}_g^\star\|^2 - \|\mathbf{u}_{g,n+1} - \mathbf{u}_g^\star\|^2 \big) + \|\mathbf{x}_{g,n} - \mathbf{x}^\star\|^2 - \|\mathbf{x}_{g,n+1} - \mathbf{x}^\star\|^2 \\
- \|\mathbf{x}_{g,n} - \mathbf{x}_{f,n}\|^2 + \chi_{4,n}.
\end{aligned}
$$

Denote $\mathbf{u}_f^\star \in \partial f(\mathbf{x}^\star)$. We have $\langle \mathbf{x}_{f,n} - \mathbf{x}^\star \mid \mathbf{u}_{f,n} - \mathbf{u}_f^\star \rangle \geq 0$, since $f$ is convex. Hence,

$$
\begin{aligned}
\chi_n &= 2\gamma_n \big( \langle \mathbf{x}_{f,n} - \mathbf{x}^\star \mid \mathbf{u}_{f,n} - \mathbf{u}_f^\star \rangle + \langle \mathbf{x}_{f,n} - \mathbf{x}^\star \mid \mathbf{u}_f^\star + \mathbf{r}_n \rangle + \langle \mathbf{x}_{g,n+1} - \mathbf{x}^\star \mid \mathbf{u}_{g,n+1} \rangle \big) \\
&\geq 2\gamma_n \big( \langle \mathbf{x}_{f,n} - \mathbf{x}^\star \mid \mathbf{u}_f^\star + \mathbf{r}_n \rangle + \langle \mathbf{x}_{g,n+1} - \mathbf{x}^\star \mid \mathbf{u}_{g,n+1} \rangle \big) \\
&= 2\gamma_n \big( \langle \mathbf{x}_{f,n} - \mathbf{x}^\star \mid \mathbf{u}_f^\star + \mathbf{r}_n \rangle + \langle \mathbf{x}_{g,n+1} - \mathbf{x}^\star \mid \mathbf{u}_g^\star \rangle + \langle \mathbf{x}_{g,n+1} - \mathbf{x}^\star \mid \mathbf{u}_{g,n+1} - \mathbf{u}_g^\star \rangle \big) \\
&\geq 2\gamma_n \big( \langle \mathbf{x}_{f,n} - \mathbf{x}^\star \mid \mathbf{u}_f^\star + \mathbf{r}_n \rangle + \mu_g \|\mathbf{x}_{g,n+1} - \mathbf{x}^\star\|^2 + \langle \mathbf{x}_{g,n+1} - \mathbf{x}^\star \mid \mathbf{u}_g^\star \rangle \big),
\end{aligned}
\tag{8}
$$

where the last inequality follows from the assumption that $g$ is $\mu_g$-strongly convex. Set

$$
\mathbf{x}_{f,n}^e = \mathrm{prox}_{\gamma_n f} \big( (\mathbf{x}_{g,n} - \gamma_n \mathbf{u}_{g,n} - \gamma_n \nabla h(\mathbf{x}_{g,n})) \big).
$$

Then, using the non-expansiveness of $\mathrm{prox}_{\gamma_n f}$, we get

$$
\|\mathbf{x}_{f,n}^e - \mathbf{x}_{f,n}\| \leq \gamma_n \|\nabla h(\mathbf{x}_{g,n}) - \mathbf{r}_n\|.
$$

Now, let us define

$$
\begin{cases}
\chi_{5,n} = \langle \mathbf{x}_{f,n} - \mathbf{x}_{f,n}^e \mid \mathbf{r}_n - \nabla h(\mathbf{x}_{g,n}) \rangle \\
\chi_{6,n} = \langle \mathbf{x}_{f,n}^e - \mathbf{x}^\star \mid \mathbf{r}_n - \nabla h(\mathbf{x}_{g,n}) \rangle \\
\chi_{7,n} = \chi_{5,n} + \chi_{6,n} = \langle \mathbf{x}_{f,n} - \mathbf{x}^\star \mid \mathbf{r}_n - \nabla h(\mathbf{x}_{g,n}) \rangle.
\end{cases}
$$

Then, we have

$$\chi_{5,n} = \left\langle \mathbf{x}_{f,n} - \mathbf{x}_{f,n}^e \mid \mathbf{r}_n - \nabla h(\mathbf{x}_{g,n}) \right\rangle$$
$$\leq \|\mathbf{x}_{f,n} - \mathbf{x}_{f,n}^e\| \cdot \|\mathbf{r}_n - \nabla h(\mathbf{x}_{g,n})\|$$
$$\leq \gamma_n \|\mathbf{r}_n - \nabla h(\mathbf{x}_{g,n})\|^2,$$

and since $\mathbf{x}_{f,n}^e$ is $\mathcal{F}_{n-1}$-measurable (by induction), we obtain

$$\mathbf{E}[\chi_{6,n}|\mathcal{F}_{n-1}] = \left\langle \mathbf{x}_{f,n}^e - \mathbf{x}^\star \mid \mathbf{E}[\mathbf{r}_n - \nabla h(\mathbf{x}_{g,n})|\mathcal{F}_{n-1}] \right\rangle = 0.$$

Furthermore, for any $\eta \in ]0,1[$, since $h$ is $\mu_h$-strongly convex and has $L$-Lipschitz continuous gradient, we have

$$2 \left\langle \mathbf{x}_{f,n} - \mathbf{x}^\star \mid \mathbf{r}_n \right\rangle = 2 \left\langle \mathbf{x}_{f,n} - \mathbf{x}^\star \mid \nabla h(\mathbf{x}_{g,n}) \right\rangle + 2 \left\langle \mathbf{x}_{f,n} - \mathbf{x}^\star \mid \mathbf{r}_n - \nabla h(\mathbf{x}_{g,n}) \right\rangle$$
$$= 2 \left\langle \mathbf{x}_{f,n} - \mathbf{x}_{g,n} \mid \nabla h(\mathbf{x}_{g,n}) - \nabla h(\mathbf{x}^\star) \right\rangle + 2 \left\langle \mathbf{x}_{g,n} - \mathbf{x}^\star \mid \nabla h(\mathbf{x}_{g,n}) - \nabla h(\mathbf{x}^\star) \right\rangle$$
$$+ 2 \left\langle \mathbf{x}_{f,n} - \mathbf{x}^\star \mid \nabla h(\mathbf{x}^\star) \right\rangle + 2\chi_{7,n}$$
$$\geq \frac{-L}{2(1-\eta)} \|\mathbf{x}_{f,n} - \mathbf{x}_{g,n}\|^2 - 2\frac{(1-\eta)}{L} \|\nabla h(\mathbf{x}_{g,n}) - \nabla h(\mathbf{x}^\star)\|^2 + 2\eta\mu_h \|\mathbf{x}_{g,n} - \mathbf{x}^\star\|^2$$
$$+ 2\chi_{7,n} + 2\frac{(1-\eta)}{L} \|\nabla h(\mathbf{x}_{g,n}) - \nabla h(\mathbf{x}^\star)\|^2 + 2 \left\langle \mathbf{x}_{f,n} - \mathbf{x}^\star \mid \nabla h(\mathbf{x}^\star) \right\rangle$$
$$\geq \frac{-L}{2(1-\eta)} \|\mathbf{x}_{f,n} - \mathbf{x}_{g,n}\|^2 + 2\eta\mu_h \|\mathbf{x}_{g,n} - \mathbf{x}^\star\|^2 + 2\chi_{7,n} + 2 \left\langle \mathbf{x}_{f,n} - \mathbf{x}^\star \mid \nabla h(\mathbf{x}^\star) \right\rangle. \quad (9)$$

Now, inserting (9) into (8), we arrive at

$$\chi_n \geq 2\gamma_n \left\langle \mathbf{x}_{f,n} - \mathbf{x}^\star \mid \mathbf{u}_f^\star + \nabla h(\mathbf{x}^\star) \right\rangle + 2\gamma_n \left\langle \mathbf{x}_{g,n+1} - \mathbf{x}^\star \mid \mathbf{u}_g^\star \right\rangle + 2\gamma_n\chi_{7,n}$$
$$+ 2\eta\mu_h\gamma_n \|\mathbf{x}_{g,n} - \mathbf{x}^\star\|^2 + 2\mu_g\gamma_n \|\mathbf{x}_{g,n+1} - \mathbf{x}^\star\|^2 - \frac{\gamma_n L}{2(1-\eta)} \|\mathbf{x}_{f,n} - \mathbf{x}_{g,n}\|^2, \quad (10)$$

since it follows that

$$2\gamma_n \left\langle \mathbf{x}_{f,n} - \mathbf{x}^\star \mid \mathbf{u}_f^\star + \nabla h(\mathbf{x}^\star) \right\rangle + 2\gamma_n \left\langle \mathbf{x}_{g,n+1} - \mathbf{x}^\star \mid \mathbf{u}_g^\star \right\rangle) - \chi_{4,n}$$
$$= 2\gamma_n(\left\langle \mathbf{x}_{f,n} - \mathbf{x}^\star \mid \mathbf{u}_f^\star + \mathbf{u}_g^\star + \nabla h(\mathbf{x}^\star) \right\rangle$$
$$= 0.$$

We derive from (10) and (8) that

$$(1 + 2\gamma_n\mu_g) \|\mathbf{x}_{g,n+1} - \mathbf{x}^\star\|^2 + \gamma_n^2 \|\mathbf{u}_{g,n+1} - \mathbf{x}^\star\|^2 + (1 - \frac{\gamma_n L}{2(1-\eta)}) \|\mathbf{x}_{f,n} - \mathbf{x}_{g,n}\|^2$$
$$\leq (1 - 2\gamma_n\mu_h\eta) \|\mathbf{x}_{g,n} - \mathbf{x}^\star\|^2 + \gamma_n^2 \|\mathbf{u}_{g,n} - \mathbf{x}^\star\|^2 - 2\gamma_n\chi_{7,n}.$$

Since $(\gamma_n)_{n\in\mathbb{N}}$ is a nonnegative sequence that converges to 0, there exists some positive integer $n_0$ such that $(1 - \frac{\gamma_n L}{2(1-\eta)}) \geq 0$ for any $n \geq n_0$. Hence,

$$(1 + 2\gamma_n\mu_g) \|\mathbf{x}_{g,n+1} - \mathbf{x}^\star\|^2 + \gamma_n^2 \|\mathbf{u}_{g,n+1} - \mathbf{x}^\star\|^2$$
$$\leq (1 - 2\gamma_n\mu_h\eta) \|\mathbf{x}_{g,n} - \mathbf{x}^\star\|^2 + \gamma_n^2 \|\mathbf{u}_{g,n} - \mathbf{x}^\star\|^2 - 2\gamma_n\chi_{7,n}, \qquad \forall n \geq n_0.$$

Now, taking the conditonal expectation with respect to $\mathcal{F}_{n-1}$, we obtain

$$(1 + 2\gamma_n\mu_g)\mathbf{E}[\|\mathbf{x}_{g,n+1} - \mathbf{x}^\star\|^2|\mathcal{F}_{n-1}] + \gamma_n^2\mathbf{E}[\|\mathbf{u}_{g,n+1} - \mathbf{x}^\star\|^2|\mathcal{F}_{n-1}]$$
$$\leq (1 - 2\gamma_n\mu_h\eta) \|\mathbf{x}_{g,n} - \mathbf{x}^\star\|^2 + \gamma_n^2 \|\mathbf{u}_{g,n} - \mathbf{x}^\star\|^2 - 2\gamma_n\mathbf{E}[\chi_{7,n}|\mathcal{F}_{n-1}] \qquad (11)$$
$$= (1 - 2\gamma_n\mu_h\eta) \|\mathbf{x}_{g,n} - \mathbf{x}^\star\|^2 + \gamma_n^2 \|\mathbf{u}_{g,n} - \mathbf{x}^\star\|^2 - 2\gamma_n\mathbf{E}[\chi_{5,n}|\mathcal{F}_{n-1}]$$
$$\leq (1 - 2\gamma_n\mu_h\eta) \|\mathbf{x}_{g,n} - \mathbf{x}^\star\|^2 + \gamma_n^2 \|\mathbf{u}_{g,n} - \mathbf{x}^\star\|^2 + 2\gamma_n^2\mathbf{E}[\|\mathbf{r}_n - \nabla h(\mathbf{x}_{g,n})\|^2|\mathcal{F}_{n-1}], \quad \forall n \geq n_0.$$

As indicated in the proof of [11], we have

$$\gamma_n^{-2}(1 + 2\gamma_n\mu_g) = \gamma_{n+1}^{-2}(1 - 2\gamma_{n+1}\mu_h\eta).$$

and

$$\lim_{n\to\infty} (n+1)\gamma_n = (\eta\mu_h + \mu_g)^{-1}. \qquad (12)$$

Therefore, by dividing both sides of (11) by $\gamma_n^2$, and taking the expectations, we obtain

$$\gamma_{n+1}^{-2}(1 - 2\gamma_{n+1}\mu_h\eta)\mathbf{E}[\|\mathbf{x}_{g,n+1} - \mathbf{x}^\star\|^2] + \mathbf{E}[\|\mathbf{u}_{g,n+1} - \mathbf{x}^\star\|^2]$$
$$\leq \gamma_n^{-2}(1 - 2\gamma_n\mu_h\eta)\mathbf{E}[\|\mathbf{x}_{g,n} - \mathbf{x}^\star\|^2] + \mathbf{E}[\|\mathbf{u}_{g,n} - \mathbf{x}^\star\|^2] + 2\mathbf{E}[\|\mathbf{r}_n - \nabla h(\mathbf{x}_{g,n})\|^2].$$

Now, summing this inequality from $n = n_0$ to $n = N$, we get

$$\gamma_{N+1}^{-2}(1 - 2\gamma_{N+1}\mu_h\eta)\mathbf{E}[\|\mathbf{x}_{g,N+1} - \mathbf{x}^\star\|^2] \tag{13}$$

$$\leq \gamma_{n_0}^{-2}(1 - 2\gamma_{n_0}\mu_h\eta)\mathbf{E}[\|\mathbf{x}_{g,n_0} - \mathbf{x}^\star\|^2] + \mathbf{E}[\|\mathbf{u}_{g,n_0} - \mathbf{x}^\star\|^2] + \sum_{k=n_0}^{N}\mathbf{E}[\|\mathbf{r}_k - \nabla h(\mathbf{x}_{g,k})\|^2].$$

In view of (12), (4) follows from (13). $\qquad\square$

We now present the key lemma for the proof of Theorem 2. This lemma is a direct corollary from [26, Lemma 4.4], hence we omit the proof.

**Lemma 1** *Let $\alpha \in ]0, 1]$, let $c$ and $\tau$ be in $]0, +\infty[$, and let $n_0$ be a positive integer. Let $(\theta_n)_{n\in\mathbb{N}}$ be a positive sequence defined by $\theta_n = cn^{-\alpha}$. Let $(s_n)_{n\in\mathbb{N}}$ be a sequence that satisfies*

$$0 \leq s_{n+1} \leq (1 - \theta_n)s_n + \tau\theta_n^2, \qquad \forall n \geq n_0.$$

*Then, $s_n$ satisfies*

$$s_n = \begin{cases} \mathcal{O}\big(1/n^\alpha\big) & \text{if } 0 < \alpha < 1 \\ \mathcal{O}\big(1/n^c\big) & \text{if } \alpha = 1, \text{ and } 0 < c < 1 \\ \mathcal{O}\big((\log n)/n\big) & \text{if } \alpha = 1, \text{ and } c = 1 \\ \mathcal{O}\big(1/n\big) & \text{if } \alpha = 1, \text{ and } c > 1. \end{cases}$$

**Proof of Theorem 2**. Taking the expectations of both sides in (11), we get

$$\mathbf{E}[\|\mathbf{x}_{g,n+1} - \mathbf{x}^\star\|^2] \tag{14}$$
$$\leq (1 + 2\gamma_n\mu_g)\mathbf{E}[\|\mathbf{x}_{g,n+1} - \mathbf{x}^\star\|^2] + \gamma_n^2\mathbf{E}[\|\mathbf{u}_{g,n+1} - \mathbf{x}^\star\|^2]$$
$$\leq (1 - 2\gamma_n\mu_h\eta)\mathbf{E}[\|\mathbf{x}_{g,n} - \mathbf{x}^\star\|^2] + \gamma_n^2\mathbf{E}[\|\mathbf{u}_{g,n} - \mathbf{x}^\star\|^2] + 2\gamma_n^2\mathbf{E}[\|\mathbf{r}_n - \nabla h(\mathbf{x}_{g,n})\|^2], \quad \forall n \geq n_0.$$

Since the learning rate $\gamma_n = \Theta(n^{-\alpha})$, we can find two positive real numbers $c_0 \leq c_1$ and a positive integer $n_1 \geq n_0$, such that $c_0 n^{-\alpha} \leq \gamma_n \leq c_1 n^{-\alpha}$ for any $n \geq n_1$. Then, we obtain

$$\mathbf{E}[\|\mathbf{x}_{g,n+1} - \mathbf{x}^\star\|^2] \leq (1 - 2\mu_h\eta c_0 n^{-\alpha})\mathbf{E}[\|\mathbf{x}_{g,n} - \mathbf{x}^\star\|^2]$$
$$+ (c_1 n^{-\alpha})^2\big(\mathbf{E}[\|\mathbf{u}_{g,n} - \mathbf{x}^\star\|^2] + 2\mathbf{E}[\|\mathbf{r}_n - \nabla h(\mathbf{x}_{g,n})\|^2]\big), \quad \forall n \geq n_1.$$

$\mathbf{E}[\|\mathbf{r}_n - \nabla h(\mathbf{x}_{g,n})\|^2]$ and $\mathbf{E}[\|\mathbf{u}_{g,n} - \mathbf{x}^\star\|^2]$ are uniformly bounded by some positive constants by assumption. Denote these constants by $\tau_0$ and $\tau_1$, then we have

$$\mathbf{E}[\|\mathbf{x}_{g,n+1} - \mathbf{x}^\star\|^2] \leq (1 - 2\mu_h\eta c_0 n^{-\alpha})\mathbf{E}[\|\mathbf{x}_{g,n} - \mathbf{x}^\star\|^2] + (2\tau_0 + \tau_1)(c_1 n^{-\alpha})^2, \quad \forall n \geq n_1.$$

Setting $\theta_n = 2\mu_h\eta c_0 n^{-\alpha}$ and $\tau = c_1^2(2\tau_0 + \tau_1)(2\mu_h\eta c_0)^{-2}$, we get

$$\mathbf{E}[\|\mathbf{x}_{g,n+1} - \mathbf{x}^\star\|^2] \leq (1 - \theta_n)\mathbf{E}[\|\mathbf{x}_{g,n} - \mathbf{x}^\star\|^2] + \tau\theta_n^2, \quad \forall n \geq n_1.$$

Proof follows from Lemma 1. $\qquad\square$