[Reviews · NeurIPS 2016]

Reviewer 1

Summary

The paper proposes an extension of the three operator splitting algorithm in [11] to a stochastic setting, to solve composite problems that are the sum of a convex function that has a Lipschitz continuous gradient and two propoer closed convex simple functions. To show convergence in expectation, additional (restricted) strong convexity assumptions are also needed.

Qualitative Assessment

1) I think it would be wise and fair to discuss the work of Raguet et al. (2013 in SIIMS) on the generalized forward-backward (GFB), and that of Bricenos-Arias (2014 in Optimization) on forward-bakcward-Douglas-Rachford (FB-DRS). These algorithms are prior to [11] and solve n even more general version of (1) in the deterministic case. 2) Extending (1) to the case of arbitrary number of non-smooth terms can be achieved easily through a product-space trick. It would be interesting to mention this in the paper.

Confidence in this Review

3-Expert (read the paper in detail, know the area, quite certain of my opinion)


Reviewer 2

Summary

In this paper the authors proposed a stochastic optimization algorithm, STCM, for a three composite convex minimization problems. This problem can be write the sum of two proper, lower semicontinuous convex functions and a smooth function with restricted strong convexity. This work is based on the deterministic three operator splitting method proposed by Davis and Yin. The author(s). The almost surely convergence and a convergence rate are established. Major comments: (1) The main result is quite clear, but lacks the support in details for important aspects. For example, it would be better to show some intuition of the proposed algorithm. If it is an directly extension of Davis and Yin’s work by changing the gradient into the stochastic gradient, authors should point out it. I would like to suggest authors to show more insights and intuition behind of the proposed algorithm, since you have enough space. The proof looks quite rush. Many derivation details are omitted. It is hard to check the correctness in proofs, although the results sound correct. (2) In the Numerical experiments section, some basic algorithms should be included for comparison, e.g., Wang et al., Random Multi-Constraint Projection: Stochastic Gradient Methods for Convex Optimization with Many Constraints, 2015. (3) It is important to compare the convergence rates with some other stochastic gradient based method. Basically, we want to know how tight it is? For example, is it consistent the standard SGD, accelerated SGD, and the method above. Otherwise, it is difficult to evaluate the theoretical merit of this work. Minor comments: (1) There are lots of typos in this paper, e.g., equation (4), the citation in line 136, line 72, line 95, line 285. (2) In line 2 of equation (10), the grad(x_g,n - x*) should be (grad(x_g,n) - grad(x*)) instead. (3) $\mu$ strong convex => $\mu$ strongly convex.

Qualitative Assessment

Major comments: (1) The main result is quite clear, but lacks the support in details for important aspects. For example, it would be better to show some intuition of the proposed algorithm. If it is an directly extension of Davis and Yin’s work by changing the gradient into the stochastic gradient, authors should point out it. I would like to suggest authors to show more insights and intuition behind of the proposed algorithm, since you have enough space. The proof looks quite rush. Many derivation details are omitted. It is hard to check the correctness in proofs, although the results sound correct. (2) In the Numerical experiments section, some basic algorithms should be included for comparison, e.g., Wang et al., Random Multi-Constraint Projection: Stochastic Gradient Methods for Convex Optimization with Many Constraints, 2015. (3) It is important to compare the convergence rates with some other stochastic gradient based method. Basically, we want to know how tight it is? For example, is it consistent the standard SGD, accelerated SGD, and the method above. Otherwise, it is difficult to evaluate the theoretical merit of this work.

Confidence in this Review

2-Confident (read it all; understood it all reasonably well)


Reviewer 3

Summary

The authors propose a new algorithm for minimizing a stochastic composite objective with three convex components. The paper includes proof of convergence and numerical experiments in the context of portfolio optimization and classification via non-linear support vector machines.

Qualitative Assessment

The topic is interesting and certainly has potential for impact. The paper is generally well written and clearly presented. However, the numerical experiments and comparison to existing methods fall short of satisfaction.

Confidence in this Review

2-Confident (read it all; understood it all reasonably well)


Reviewer 4

Summary

In this paper, the authors a stochastic optimization method for the convex minimization of the sum of three convex functions. They prove the convergence characterization of the proposed algorithm. Finally, some numerical experiments are conducted to show the the effectiveness of the proposed method.

Qualitative Assessment

(1) This paper combine the "three operator splitting method" with "stochastic analysis". In fact, the main proof techniques are standard. Hence, I did not find the results very exciting. (2) The author claim that their method is the first purely primal stochastic splitting method which uses the proximity operator f and g separately, which is the main contribution. In fact, the literature "Accelerated stochastic gradient method for composite regularization (AISTATS), L.W. Zhong, J.T. Kwok" has proposed primal method for solving this class minimizations. (3) The numerical experiments are conducted unfairable. They compared their algorithm against the standard deterministic three-operator splitting method in [11] with a low accuracy. The authors should conducts more experiments to shown the efficiency of their algorithm by comparing with some state-of-the-art method such as the methods in " "Accelerated stochastic gradient method for composite regularization (AISTATS)".

Confidence in this Review

3-Expert (read the paper in detail, know the area, quite certain of my opinion)


Reviewer 5

Summary

This paper proposed an algorithm for the convex minimization of the sum of three convex functions, one of which has Lipschitz continuous gradient as well as restricted strong convexity. The convergence rate in expectation and the almost sure convergence of the algorithm were derived.

Qualitative Assessment

Stochastic gradient method for solving composite convex minimization with one regularizer was studied in the following papers: Lan. An optimal method for stochastic composite optimization, 2012. Ghadimi, Lan. Optimal Stochastic Approximation Algorithms for Strongly Convex Stochastic Composite Optimization, 2012. In the current paper, the authors did not highlight the main difficulty in solving composite convex minimization with two regularizers. Moreover, the authors did not explain how to calculate the variance of the gradient vector in Example 5.2.

Confidence in this Review

2-Confident (read it all; understood it all reasonably well)